# Are Cell-Based Therapies Safe and Effective in the Treatment of Neurodegenerative Diseases? A Systematic Review with Meta-Analysis

**DOI:** 10.3390/biom12020340

**Published:** 2022-02-21

**Authors:** Jasper Van den Bos, Yousra El Ouaamari, Kristien Wouters, Nathalie Cools, Inez Wens

**Affiliations:** 1Laboratory of Experimental Hematology, Vaccine and Infectious Disease Institute (Vaxinfectio), University of Antwerp, Universiteitsplein 1, B-2610 Antwerpen, Belgium; yousra.elouaamari@uantwerpen.be (Y.E.O.); nathalie.cools@uantwerpen.be (N.C.); inez.wens@uza.be (I.W.); 2Clinical Trial Center (CTC), CRC Antwerp, Antwerp University Hospital, University of Antwerp, Drie Eikenstraat 655, B-2650 Edegem, Belgium; kristien.wouters@uza.be; 3Center for Cell Therapy and Regenerative Medicine (CCRG), Antwerp University Hospital, Drie Eikenstraat 655, B-2650 Edegem, Belgium

**Keywords:** regeneration, cell-based therapy, neurodegenerative diseases

## Abstract

Over the past two decades, significant advances have been made in the field of regenerative medicine. However, despite being of the utmost clinical urgency, there remains a paucity of therapeutic strategies for conditions with substantial neurodegeneration such as (progressive) multiple sclerosis (MS), spinal cord injury (SCI), Parkinson’s disease (PD) and Alzheimer’s disease (AD). Different cell types, such as mesenchymal stromal cells (MSC), neuronal stem cells (NSC), olfactory ensheathing cells (OEC), neurons and a variety of others, already demonstrated safety and regenerative or neuroprotective properties in the central nervous system during the preclinical phase. As a result of these promising findings, in recent years, these necessary types of cell therapies have been intensively tested in clinical trials to establish whether these results could be confirmed in patients. However, extensive research is still needed regarding elucidating the exact mechanism of action, possible immune rejection, functionality and survival of the administered cells, dose, frequency and administration route. To summarize the current state of knowledge, we conducted a systematic review with meta-analysis. A total of 27,043 records were reviewed by two independent assessors and 71 records were included in the final quantitative analysis. These results show that the overall frequency of serious adverse events was low: 0.03 (95% CI: 0.01–0.08). In addition, several trials in MS and SCI reported efficacy data, demonstrating some promising results on clinical outcomes. All randomized controlled studies were at a low risk of bias due to appropriate blinding of the treatment, including assessors and patients. In conclusion, cell-based therapies in neurodegenerative disease are safe and feasible while showing promising clinical improvements. Nevertheless, given their high heterogeneity, the results require a cautious approach. We advocate for the harmonization of study protocols of trials investigating cell-based therapies in neurodegenerative diseases, adverse event reporting and investigation of clinical outcomes.

## 1. Introduction

Over the past two decades, significant advances have been made in the field of regenerative medicine, which focuses on developing methods to restore, regrow or replace damaged or dysfunctional cells, tissues and organs [1]. In 2009, Chondroselect (TiGenix N.V.) was the first regenerative cellular therapy that entered the European market, using autologous ex vivo expanded chondrocytes. Ever since, dozens of regenerative treatments have been successfully launched on the market targeting, amongst other things, cartilage defects, burn wounds, osteoarthritis, Crohn’s disease and ischemia [1]. Despite promising innovations in this medical field, treatment strategies to prevent neuronal loss in neurodegenerative diseases such as (progressive) multiple sclerosis (MS), spinal cord injury (SCI), Parkinson’s SP disease (PD) and Alzheimer’s SP disease (AD), which are characterized by a progressive loss of structure and function of neurons in the central nervous system, are currently lacking [2]. For these conditions, disease-modifying treatments or therapies, which alleviate symptoms but do not slow the disease progression, are primarily available. While substantial progress has been made with these drugs, efficacy often comes at a price, inducing damaging side-effects due to the generalized immune modulation of these therapies [3,4,5]. Therefore, new strategies to derive neuronal subtypes and to create environmental changes promoting neuroprotection and the reconstruction of a neuronal network are of utmost clinical urgency [6].

Different cell types, such as mesenchymal stromal cells (MSC) [7,8,9], bone-marrow mononuclear cells (BMMC) [10,11], olfactory mucosa auto grafts (OMA) [12], olfactory ensheathing cells (OEC) [13,14], neuronal stem cells (NSC) [15], Schwann cells [16] and macrophages [17], showed safety and promising regenerative properties. Several studies reported recovery in experimental autoimmune encephalomyelitis (EAE), which is an MS mouse model, including promotion of neuronal repair, neuroprotection and decreased demyelination [9,18]. Moreover, various cell types in SCI rat models showed recovery after complete SCI after transplantation. In addition to neuroprotective mechanisms, axon regeneration and remyelination were observed, leading to significant sensory and motor improvements [8,13,15,16]. In addition, different types of neuronal dopaminergic grafts showed survival and improved functionality in PD rat models [19,20,21]. As a result of these promising findings, in recent years, different types of cell therapies have been intensively investigated in clinical trials to establish whether these positive results could be confirmed in humans. In order to summarize the current state of knowledge, we conducted a systematic review and meta-analysis. In this article, we will discuss the most frequently used cell therapies for neurodegenerative diseases and examine whether the treatments are safe and exhibit improvement in clinical outcomes.

## 2. Methods

### 2.1. Literature Search

The protocol of the present systematic review was registered on PROSPERO (CRD42021288757).Two independent reviewers (JVdB and IW) systematically performed the literature search in the electronic databases PubMed, Web of Science and Clinicaltrials.gov. The final search was performed on the 2nd of December 2021. The search terms used for the search on PubMed were (“Nerve Regeneration”[MeSH Terms] OR “Nerve Regeneration”[Title/Abstract] OR “Biological Therapy”[MeSH Terms]) AND ((“multiple sclerosis”[Title/Abstract] OR “spinal cord injury”[Title/Abstract] OR “Parkinson disease”[Title/Abstract] OR “Alzheimer disease”[Title/Abstract]) OR (“Parkinsonian Disorders”[Mesh] OR “Multiple Sclerosis”[Mesh] OR “Spinal Cord Injuries”[Mesh] OR “Alzheimer Disease”[Mesh])). The search terms used for the search on clinicaltrails.gov were (Multiple sclerosis/Parkinson disease/Alzheimer disease /Spinal cord injury) AND (Regeneration/Remyelination/Biological therapy). The same terms were used for Web of Science, excluding ‘biological therapy’, due to the fact that mostly irrelevant search results were obtained. The following filters were applied in the databases PubMed and Web of Science in order to specify the search: (1) publication date between 1 January 2000 and 2 December 2021, (2) clinical trials, and (3) studies performed on subjects older than 18 years.

### 2.2. In- and Exclusion Criteria

Clinical trials reporting on treatment interventions in adult persons with neurodegenerative diseases including MS, AD, PD and SCI were included. The interventions studied were human cellular therapies inducing regeneration, including MSC, BMMC, peripheral blood derived stem cells (PBSCs), macrophages, NSC, neuron transplantations, OEC and OMA. Trials investigating chemical compounds, non-cellular biologicals and non-human cells as well as clinical trials primarily focusing on immunomodulatory effects were excluded. Since this is a quite recently emerging field, all types of studies including phase I, II and III, controlled and uncontrolled, randomized and non-randomized, blinded and unblinded studies were included.

### 2.3. Outcome Measures

The primary outcome was safety, including the proportion and number of patients with at least one serious adverse event (SAE) or at least one adverse event (AE) or with at least one SAE related to the intervention. The secondary outcome was efficacy. In studies with MS patients, efficacy was assessed by the proportion of patients with an improvement in the expanded disability status scale (EDSS). In studies with SCI patients, efficacy was assessed by the proportion of patients with an improvement in the American spinal injury association impairment scale (AIS). No analysis was performed for efficacy in studies for PD and AD, due to an inadequate number of studies with clinical data for these diseases. In addition, to the extent possible, a comparative analysis for efficacy was performed with respect to the control group. Finally, if possible, a comparative analysis was performed regarding administration route, including a comparison between intravenous (IV) administration and intrathecal (IT) administration.

### 2.4. Selection Process and Data Extraction

Two review authors (JVdB and IW) conducted the literature search separately, screened studies for eligibility and extracted the data. Disagreements between the two reviewers were clarified and solved by a consensus meeting. The initial selection was determined based on title, abstract, and keywords. Afterwards, the assessors checked for duplicates, using Doi and NCT number, which were subsequently excluded. A final selection of studies was made after reading the full text. When provided, the following information was extracted from the studies: (1) general information including Doi, NCT number, author, title and publication date; (2) study characteristics including type of neurodegenerative disease, cell type used, study design, duration follow-up and sample size; (3) participant characteristics including, gender, age, participants recruited and dropout rate; (4) information about the intervention including route of administration, dose and frequency and autologous versus allogenic administration; and (5) outcomes including the number of patients in the controlled/treated group, adverse events (causality and severity) and improvement in clinical outcomes such as AIS grade and EDSS.

### 2.5. Study Risk of Bias Assessment

Risk of bias assessment varied according to the study design. For randomized studies, the Risk of Bias (Rob2) tool from Cochrane was used [22]. For non-randomized controlled and uncontrolled studies, the ROBINS-I tool was used [23]. Assessments were performed at the study level. Risk of bias (RoB) was ranked as low risk, some concern or high risk and was visualized with the ROBVIS tool [24]. Disagreement between the two review authors (JVdB and IW) was resolved by a consensus meeting.

### 2.6. Data Synthesis and Statistical Analysis

The PRISMA guidelines were used in the implementation of the entire literature review process [25]. Figure 1 shows a flow diagram reporting the number of in- and excluded studies and the reason for exclusion. For the meta-analysis, categorical data were summarized as numbers or percentages. A random effects meta-analysis was used to calculate overall proportions and corresponding 95% confidence intervals. Studies with a control arm were included in a comparative random effects meta-analysis. Risk ratios and 95% confidence intervals are reported. For rare outcomes (such as serious adverse events), risk differences were used as effect measure instead, as risk ratios cannot be computed when no events are observed in the and control arm. All analyses were performed in R version 3.6 or higher.

## 3. Results

### 3.1. Seacrch Results

The results of the search strategy are shown in a PRISMA flow diagram (Figure 1). In total, 27,043 records were screened and sixty-one were ultimately included in the quantitative analysis. Ten additional papers, fulfilling the inclusion criteria, were included in the analysis during the literature review and after consensus between the two assessors. The majority, namely 42 of the studies, evaluated the safety and/or efficacy of the administration of stem cells including MSC. Twelve studies regarding BMMCs, PBSCs and macrophages were included. The seventeen remaining studies investigated the administration or transplantation of neuronal cells including NSC, OMA, OEC, Schwann cells and neuron-like cell types (Table 1).

### 3.2. Mesenchymal Stromal Cells Exert Neuroprotective and Immunomodulatory Properties after Transplantation

Currently, most of the studies that are being investigated in clinical trials are focusing on cell-based therapies using MSCs. These cells are characterized as pluripotent stem cells that can differentiate into osteoblasts, chondrocytes and adipocytes [26]. They can be easily extracted from diverse types of tissue, including bone marrow, adipose tissue, placenta and the umbilical cord [26,27,28,29], and are characterized by their adherent properties as well as their expression of CD44, CD73, CD90 and CD105, as well as lacking the expression of CD14, CD19, CD34, CD45 and HLA-class II [30,31]. MSCs are an ideal cell source for tissue regeneration owing to their interesting neuroprotective properties. They excrete growth factors, neurotrophins and cytokines inhibiting neuronal loss through anti-apoptotic effects and inducing neurogenesis and angiogenesis creating a favourable microenvironment for remyelination or regeneration [32,33]. Moreover, they exert immunomodulatory properties and more specifically create an anti-inflammatory environment by interacting with a variety of immune cells, such as the induction of inhibitory phenotypes in antigen presenting cells (APC), the downregulation of natural killer cells and cytotoxic T lymphocytes, driving the differentiation of naïve CD4^+^ lymphocytes into T helper (Th) cells and inducing an increase in T cells differentiating in CD4^+^ CD25^+^ regulatory T cells (Tregs) [34]. Altogether, these features make MSCs an interesting candidate for the treatment of neurodegenerative diseases.

Presently, 42 clinical trials have investigated the safety and/or efficacy of bone marrow-derived MSC (BMMSC), adipose-derived MSC (ADMSC), umbilical cord-derived MSC (UC-MSC) and placenta-derived MSC (PD-MSC) in MS, SCI, PD and AD. Autologous BMMSCs and ADMSCs are the most commonly used. Interestingly, ADMSCs expose superior characteristics for clinical use, as the isolation is less painful and invasive, they reach a higher yield, have better proliferative properties and demonstrate a superiority in the production of cytokines and neuroprotective factors, as compared to BMMSCs [35,36,37,38,39,40]. In addition, allogenic MSCs such as UC-MSCs and PD-MSC are also emerging in the clinical field [41,42,43,44,45]. Besides their non-invasive harvesting method, these cells possess an extremely high proliferative ability and improved therapeutic properties in comparison with ADMSCs [39,46]. The administration routes vary widely among the different clinical trials. Most protocols use IT injection, even though this technique is mildly invasive and is associated with some side effects such as transient headaches, mild fever and very exceptionally meningitis [37,47,48,49] or cerebrospinal fluid (CSF) leak [50]. However, because of a better migration to the lesion site in the central nervous system (CNS) via the CSF and a superior neuroprotective effect in preclinical models, IT injection is preferably used over IV administration [9,47,51,52,53]. In addition, other methods are used in SCI, PD or AD such as administration into the injury site and administration into the brain by surgery [43,50,54,55]. Finally, four of these 42 studies used biomaterial scaffolds in combination with UC-MSCs or BMMSCs in patients with SCI [56,57,58,59]. These scaffolds are used to bridge the lesion gap and guide axonal growth across the injury site. In addition, they can be used to deliver stem cells and functional biomolecules, such as brain-derived neurotrophic factor (BDNF) [60], optimizing the microenvironment to promote survival of the administered cells and inducing regeneration at the injury site [61,62,63]. Interestingly, no adverse events related to the surgical administration of scaffolds in combination with MSCs were reported. The study by Zhao et al. [58] showed only sensory improvements, in contrast to the other studies which showed motor and sensory improvements in the majority [56,57] or even all the SCI patients [59]. To date, there are no universal dosage guidelines for MSCs in cell-based therapy. Therefore, the number of administrations and the number of cells administered varies greatly among the different clinical trials. Nevertheless, several studies showed superior improvements in EDSS status and AIS for repeated administration of BMMSC over a single dose [9,64,65].

### 3.3. Bone-Marrow and Peripheral Blood Stem Cells Showed Clinical Improvement in Some Patients

Eight studies, seven in SCI and one in MS, using BMMC were included. This cell fraction, usually aspirated from the iliac crest, comprises a heterogeneous population of cells with a variety of functions including hematopoietic stem cells (HSCs), endothelial progenitor cells, MSCs, monocytes, and lymphocytes [66]. Several of these cells have shown neuroprotective and regenerative properties in preclinical studies, although the exact mechanisms involved remain to be defined [10,11,67,68]. BMMCs have the advantage that they do not require any isolation, cultivation or expansion steps, minimizing the risk of contamination and functional or genetic instability [69]. On the other hand, BMMCs may contain inflammatory components such as activated macrophages and lymphocytes that could inhibit the regeneration process. Nevertheless, none of these studies reported a (serious) AE suggesting this [69,70,71,72,73,74,75,76]. No universal administration and dosing guidelines are available, and therefore these parameters varied significantly between the studies. Clinical trials performed by Rice et al. and Syková et al. involving IV administration did not show clinical improvement in both MS and SCI patients [69,74]. In contrast, studies administering the cells close to the injury site including IT or intra-arterial (IA) administration and injection into the injured spinal cord showed marked clinical improvements in several SCI patients. [71,72,75,76]. These findings were confirmed in the study by Syková et al. comparing IV and IA administration near to the injury site in SCI [74]. Interestingly, Park et al. and Yoon et al. administered granulocyte macrophage-colony stimulating factor (GM-CSF) in addition to bone marrow cell transplantation [72,75]. This resulted in more stem cell mobilization, enhanced survival and an improved neuroprotective and regenerative effect by inducing cytokine production. There were no other AEs observed than fever due to the GM-CSF administration [72,75,77]. Finally, four studies investigated the safety and efficacy of autologous obtained stem cells of the PBSCs and macrophages in SCI patients [78,79,80,81]. PBSCs show similar properties to the BMMCs and were administered IT, IA or by injection into the spinal cord, whereas macrophages have immunomodulatory and wound healing properties possibly leading to neurologic restoration. No issues regarding safety after PBSC administration were reported and several patients showed an improvement in AIS grade [79]. In patients who received autologous macrophages, no (serious) AEs were reported related to the cellular product. Indeed, seven SAEs related to the surgical procedure, microinjections into the injured spinal cord, serve as an indication for the invasive character of this method. Both studies showed preliminary efficacy as several patients’ AIS grade improved.

### 3.4. Transplantation of Neural Cells Induces Reorganisation and Repair of the Neural Network

Over the past few years, several clinical trials investigated the application of neural transplantation to induce regeneration. Five studies used NSCs derived from the foetal brain or spinal cord [82,83,84,85,86], as these multipotent cells have the potential to self-renew and differentiate in a site-specific manner into NSC cell types including astrocytes, oligodendrocytes and neurons [87]. Despite the allogeneic origin of the NSCs, they were well-tolerated and safe under the administration of immunosuppressive drugs to prevent immunologic rejection of the stem cells which was observed in earlier clinical trials [88,89]. No tumours or abnormalities were detected after 1 year of follow-up [82,83,85,86]. One study reported safety up to 6 years after administration [84]. Nevertheless, the six SAEs related to the surgical procedure, including three CSF-leaks, one pseudo-meningocele and two Staph. epidermis infections, underline the invasive nature of the procedure. The number of administered cells, injected at the injury site in the spinal cord, ranged between 20 and 100 × 10^6^ cells. Clinical improvements were observed in two studies including the placebo-controlled study of Shin et al. [84,85], whereas only sensory changes [86] or no improvements at all were reported in the other studies [82,83].

Two studies, from the group of Lima et al. investigated the regenerative properties of OMA in SCI [90,91]. The olfactory mucosa is an interesting target for cellular therapy as it exhibits the fastest rate of neurogenesis in adults and consists of two distinct cell populations, namely the OEC and NSC. OEC are specialized glial cells who myelinate axons both in the CNS and peripheral nervous system (PNS) and exhibit a mixture of astrocyte-specific and Schwann cell-specific characteristics [92]. Preclinical studies showed potential to myelinate and promote axonal growth in the injured spinal cord [14]. This can, in combination with the properties of the previously described NSCs lead to the formation of new neuronal networks [13,14,90,91,93,94]. In addition, the olfactory mucosa is an ideal graft because it can be acquired autologously with minimal invasive techniques, allowing NSCs to integrate in a controlled manner without rejection [90]. The autografts were administered, after removal of the scar tissue, into the cavity of the injury. No AEs related to the cell therapy were observed. Nevertheless, due to the invasive nature of the administration method, one SAE directly related to the procedure was reported, namely meningitis [90]. Lima et al. reported several patients with improvements in the AIS scale as well as in motor, sensory and urodynamic parameters [90,91]. Four studies investigated the safety and/or efficacy of autologous or allogenic OEC in patients with SCI which were administered into the injured spinal cord after laminectomy [95,96,97,98]. Safety and feasibility were reported in all trials [95,97,98], and one study reported safety during a period up to three years post transplantation [96]. The small placebo-controlled study from Tabakow et al. reported modest improvements in sensory and motor function in comparison to the control group [97], whereas the study of Wu et al. only showed slight sensory improvements and the absence of motor improvements, and the study of Mackay-Sim et al. showed no clinical improvement [96,98]. Two studies investigated autologous Schwann-cell administration which has comparable properties to OEC and a similar administration procedure. Both studies reported safety and some clinical improvements in two patients with SCI [99,100]. Finally, four studies investigated the survival, safety and or efficacy of the transplantation of different cell types in patients with PD including embryonic dopamine neurons, embryonic caudate and putamen grafts, autologous carotid body glomus cells and human retinal pigment epithelial cells attached to micro carriers [89,101,102,103]. The cell products of grafts were implanted by an invasive surgical procedure into the putamen. All the clinical trials reported the survival of the transplants; however, three SAEs related to the surgical procedure were observed in the study of Minguez-Castellanos et al. [101]. In addition, improvements in the Unified Parkinson’s Disease Rating Scale (UDPRS) scores were noted in some patients. Noteworthily, in the sham-surgery controlled study of Freed et al., clinical deterioration was observed between 6 and 12 months post transplantation and was probably caused by graft-induced dyskinesia (GID) [89]. This raised serious concerns and led to the termination of implanting dopamine neurons into the brain of PD patients.

### 3.5. Quality Assessment Risk of Bias

Twelve out of thirteen randomized and placebo controlled-clinical trials were at a low risk of bias as assessed with the RoB2 tool, and only one study was at high risk due to bias in the measurement of outcomes and reporting of the data (Appendix A) [22]. Most of the non-randomized studies were at medium risk of bias using the ROBINS-I tool, which was associated with the open label treatment without masking assessors and/or patients to the intervention (Appendix B) [23]. Nine of these non-randomized studies were at high risk. Indeed, the combination of the cell-based treatment and immunosuppressive therapies in MS and Levodopa in PD can lead to a potential source of confounding. Finally, one study was at critical risk of bias due to the exclusion of more than 30% in the data reporting [104].

### 3.6. Meta-Analysis

#### 3.6.1. Serious Adverse Events

Out of the 70 studies, 39, including 10 controlled trials, reported adequate safety data involving the proportion of patients experiencing SAEs. In these, 53/883 patients on the experimental arms encountered SAEs, in 19 out of 39 studies. The overall frequency of SAEs was low: 0.03 (95% CI: 0.01–0.08). Five studies showed a significantly higher proportion of patients experiencing SAEs in comparison to the global effect of the studies (Figure 2). Forty studies were included in the calculation of the proportion of patients experiencing an SAE related to the cellular therapy or administration procedure. The overall incidence of related SAEs was low—0.02 (95% CI: 0.01–0.04)—and only one study by Minguez-Castellanos et al. [101] showed a significantly higher proportion of SAEs related to the administration regimen and/or cell product in comparison to the global effect of the studies (Appendix C).

The incidence of SAEs was compared between intervention and control groups in nine of the studies and was 0.03 (95% CI: 0.01–0.20) and 0.06 (95% CI: 0.02–0.19), respectively. The risk difference for SAEs in intervention groups versus control groups was −0.01 (95% CI: −0.07–0.05) and was not significant *p* = 0.73 (Figure 3).

#### 3.6.2. Adverse Events

Thirty-three studies, including eight controlled trials, reported adequate data regarding the proportion of patients experiencing AEs. Overall, 331/511 patients in the experimental groups experienced an AE in 32 out of 33 studies. The overall frequency of adverse events was high: 0.857 (95% CI: 0.66–0.95). The incidence of AEs between the intervention and control groups in eight studies was compared by calculating the risk-difference and risk ratio. The calculated risk difference was 0.02 (95% CI: −0.04–0.09, *p* = 0.49), and the risk ratio for AEs in intervention groups versus control groups was 1.00 (95% CI: 0.94–1.06, *p* = 0.96) (Appendix D).

#### 3.6.3. Clinical Response in MS Patients

Ten studies reported adequate results of the EDSS status in MS patients. Out of 188 patients, 52 improved their EDSS in the experimental arm. The proportion of clinically improved patients after a 12-month follow up was 0.30 (95% CI: 0.17–0.46) (Figure 4A). No comparative analysis was done because only two studies were included and the analysis would therefore have no added value. The efficacy of two different administration routes, namely IT and IV, was compared by means of a subgroup difference test. The proportion of patients who improved their EDSS status in the IT group was 0.34 (95% CI: 0.17–0.57), whereas this proportion was lower in the IV group—0.16 (95% CI: 0.10–0.26). However, the result for the subgroup differences was not significant, χ^2^ = 3.11 and *p* = 0.078 (Figure 4B).

#### 3.6.4. Clinical Response in SCI Patients

Thirty clinical trials with SCI patients were included in this analysis evaluating the proportion of patients who improved their AIS grade. Out of 687 SCI patients, 229 improved their AIS grade in the experimental arm, resulting in a proportion of clinical improved patients after a 6–12 month follow up of 0.35 (95% CI: 0.25–0.46) (Figure 5A). One study by Amr et al. showed a significantly higher proportion of improvement in comparison to the global effect of the studies, as all 14 patients changed their AIS grade [59]. On the other hand, one study showed significantly lower improvement in AIS grade, as 0 out of 25 patients showed clinical changes [105]. In addition, a risk-ratio analysis in eight studies, comparing the intervention and control group, was performed. The proportion of AIS-improved patients in the experimental group was remarkably higher (0.35 (95% CI: 0.27–0.44)) than in the control group (0.04 (95% CI: 0.01–0.22)). The calculated risk ratio was 3.89 (95% CI: 1.14–13.23, *p* = 0.030) (Figure 5B).

## 4. Discussion

This systematic review with meta-analysis aimed to evaluate the safety and efficacy of regenerative cell-based therapies in neurodegenerative diseases including MS, SCI, PD and AD. First, the majority of the studies reported that the application of cell therapy is safe and feasible in these patients. In particular, the results demonstrate that the frequency of SAE after administration of a cell-based treatment was low (0.03 (95% CI: 0.01–0.20)). In addition, no significant differences in SAE occurrence were observed between the control and treatment groups. Second, studies in MS and SCI patients demonstrated efficacy of cell-based therapies, evaluating clinical outcomes via EDSS and AIS grade, respectively. Noteworthily, the proportion of MS patients that improved their EDSS status during one year of follow up was 0.30 (95% CI: 0.17–0.46), whereas the proportion of SCI patients with a AIS grade improvement was 0.35 (95% CI: 0.25–0.46) during a 6–12-month follow up. All results should be interpreted with caution and will be discussed below.

### 4.1. Cell-Based Therapy in Neurodegenerative Diseases Showed Safety and Feasibility

In general, most of the studies reported safety and feasibility of the treatment based on the incidence of SAEs. In particular, the results demonstrate that the frequency of SAE after administration of a cell-based treatment was low (0.03 (95% CI: 0.01–0.20)). Nevertheless, five studies showed a significant higher incidence of SAEs which were related to the administration regimen rather than the use of the cell product itself. Furthermore, the incidence of AE was high (0.86 (95% CI: 0.66–0.95)) both in control and treatment groups and showed no difference in the comparative analysis. Such high incidence might be attributed to the severe clinical condition of the patients and the associated medical complications such as urinary tract infections which occur at a rate of more than two times a year in patients with SCI and advanced MS [106,107].

### 4.2. Modest Clinical Improvements Were Observed in MS Patients’ EDSS Scores

The EDSS of Kurtzke was used as an outcome to assess efficacy of the cell-based treatments, as it is the most frequently used and best-known instrument to monitor disease progression in MS [108,109]. Our analysis reported a proportion of 0.30 (95% CI: 0.17–0.46) MS patients that improved their EDSS status during a one year follow up. This represents a remarkably high proportion for a disease with gradually worsening clinical manifestation, indicating some promising efficacy. However, the included population is composed of subjects who have varying subtypes of MS and different EDSS scores at baseline, which might have an impact on the course of the disease [110]. Noteworthily, whereas most trials had an experimental design with a small sample size, Uccelli et al. included 134 (predominantly RR) MS patients in a randomized, double blinded, placebo-controlled clinical trial, administering BMMSC intravenously. However, they did not show any difference in EDSS improvement between the treatment and control groups [111].

### 4.3. AIS Grade Improved in Several SCI Patients after Cell-Based Therapy

American Spinal Injury Association Impairment Scale (AIS) was used to assess efficacy of the cell-based treatments in patients with SCI. The AIS classification has a tremendous prognostic value and defines precisely the level and degree of a patient’s deficit both on motor and sensory level [112]. Our analysis showed that the proportion of patients that improved their AIS grade was 0.35 (95% CI: 0.25–0.46) in the experimental group. In addition, a comparative risk ratio analysis showed significant higher clinical improvement in the treatment group versus control. However, these results must be assessed with caution. Apart from the heterogeneity in study design, dose, administration route and cell type, there are additional factors influencing the recovery rate, including the AIS-grade, injury-level and the time that has elapsed since injury [113]. The time elapsed since injury has a major influence on the difference in recovery rate. Acute AIS-A patients have about a 20% chance of spontaneously improving their grade [113,114], whereas only 5.6% of patients who remain AIS-A after one year improve their grade in the period up to five years after the injury [114]. The majority of patients that were included were classified with chronic and complete injury (AIS-A grade), as these patients showed the lowest rate of spontaneous recovery, allowing for better determination of the exact impact of the treatment [112]. In addition, the risk of conducting further damage as a result of the surgical procedure or the cell product was minimised.

### 4.4. Further Optimisation of Administration Regimen and Dosing Is Necessary

Despite promising results regarding efficacy in various clinical trials and proven safety and feasibility, many questions remain unanswered. The administration route, dose and frequency vary widely between the individual studies. Even among studies for the same disease and using the same cell type, there is large variability between these parameters. IV and IT administration are the least invasive procedures to administer cell products and are generally considered safe. However, two encephalopathies were observed after IT administration of BMMSCs in patients with MS, probably due to a secondary reaction after lysis of the administered cells in the CSF [47,48]. In addition, two cases of meningitis were reported in MS patients immediately after IT administration of MSCs [49]. Therefore, it is important to optimize the administration regimen in order to minimize the risk of side effects. Several preclinical studies, both in MS and SCI models, have already shown that IT administration shows better efficacy and migration than the IV route [9,47,53,115]. In addition, a comparative study of Petrou et al. showed superior clinical improvements after BMMSC administration in progressive MS patients for the IT route [116]. However, our comparative analysis in MS patients did not show significant clinical improvement for IT over IV administration, although the *p*-value (0.078) did suggest a tendency towards an increased improvement in the patients who received the cellular therapy IT. However, the number of studies in the IV group was limited (*n* = 3 versus *n* = 7 in the IT group) and the study by Uccelli et al. [111] had a substantial weight, meaning that we have to approach these results carefully.

Several clinical and preclinical studies [9] reported superior results after multiple doses and a peak in clinical improvement in patients after 1–3 months [117,118,119]. Repeated doses, as reported in the study by Vaquero et al. [120], would therefore be recommended to achieve a more beneficial clinical effect. Furthermore, in SCI, PD and AD patients, the cells were administered mainly at the site of injury to allow them to exert an optimal neuroregenerative effect. This often involved invasive and high-risk surgical procedures, including durotomy and laminectomy, together with the inevitable complications. Consequently, repeated doses are in this case a very high risk and may not balance out the benefits.

### 4.5. Several Hurdles Remain to Be Surpassed in Order to Develop a Successful Neuroregenerative Cell-Based Therapy

Indeed, all results should be interpreted with caution. Besides the already mentioned wide heterogeneity between the studies regarding cell type, administration regimen, dose and the disease status in the patient population, it should be noted that the majority of the studies compromise early-stage clinical trials which have a small sample size and a lack of blinding. In particular, the latter might induce some challenges. In case the cell product has to be administered via an invasive surgical procedure, ideally a sham-surgical control would be preferred. Nevertheless, the risk associated with these surgical procedures raises significant ethical issues, highlighting the hurdles to achieve an optimal trial design versus what is ethically correct. Furthermore, the results of many of the records found on clinical trials.gov remain unpublished, which may point towards a publication bias. In addition, it has already been shown in the studies of the NECTAR program that promising results in preclinical studies and early human clinical trials for cellular therapies including ventral mesencephalic (VM) transplants [21,121,122,123] do not necessarily guarantee further treatment success. Indeed, after performing two placebo-controlled clinical trials [88,89], transplanted neurons showed survival in the brain of the patient and some variable clinical improvements in younger patients in the first months after transplantation. The clinical trial performed by Freed et al. did not use immunosuppressive therapy, and therefore immune rejection of the grafts was able to occur, leading to the development of GID [89]. This was confirmed by the study of Olanow et al. in which patients improved in the first six months while immunosuppressants were administered, but deteriorated after the termination of the immunosuppressive therapy. These findings were further supported by the analysis of post mortem tissue showing the presence of activated immune cells such as microglia in and around the graft deposits [88]. In addition, Lewy bodies, which are abnormal deposits of the alpha-synuclein protein and PD pathology, were observed in several post mortem analyses, also pointing to biological restrictions of direct cell transplantation into the brain [124,125]. These findings highlight that several hurdles remain to be faced regarding biological restrictions on the road to regenerative cell-based therapies. Therefore, extensive research will have to be conducted in the future including elucidating the exact mechanism of action, possible immune rejection, functionality and the survival of the administered cells to draw adequate conclusions. In doing so, controlled or comparative and blinded studies with a large sample size and longer follow-up should be performed using the same cell type. In addition, post mortem analysis of the engrafted tissue that was administered at the injury site needs to be conducted to investigate eventual biological restrictions such as immune rejection.

### 4.6. Future Perspectives

Ongoing and future studies will help to define the exact mechanism of action, functionality and survival of the administered cells, the dose, treatment schedule and route of administration of cell-based therapies in neurodegenerative diseases as well as the exact mechanism of action which leads to clinical improvement and the survival and engraftment in the host tissue in patients with neurodegenerative diseases. Furthermore, research into innovative and novel therapeutic strategies needs to be pursued. Indeed, in the last few years, several trials have investigated the combination of cell administration with biomaterials such as scaffolds in patients with SCI [56,57,58,59]. These trials showed encouraging efficacy results, underlining the need to conduct research into new therapeutic strategies to enhance the microenvironment, which is a crucial factor to successfully induce the regeneration process [126,127]. In addition, the inflammatory microenvironment which is often observed in MS and SCI is unfavourable for CNS repair and can be adapted by the administration of anti-inflammatory cells such as regulatory T cells, which are currently extensively tested in the clinic [128,129,130]. These cells can be easily engineered to overexpress neuroprotective factors such as neurotrophins, thereby improving their neuroregenerative and neuroprotective properties [131,132]. Optimizing and combining different therapeutic strategies will possibly bring a solution for neurodegenerative diseases in the future.

## 5. Conclusions

Our analyses showed that cellular-based therapies are safe and feasible and showed promising clinical improvements in several patients with MS and SCI (Figure 6). However, taking into account the failure of NECTAR studies in PD patients in the past, extensive research into the exact mechanism of action, functionality, and survival of the administered cells, combinations of different therapeutic strategies, administration routes and dose regimens is still needed to successfully develop a regenerative cell-based therapy for neurodegenerative diseases.

## Figures and Tables

**Figure 1 biomolecules-12-00340-f001:**
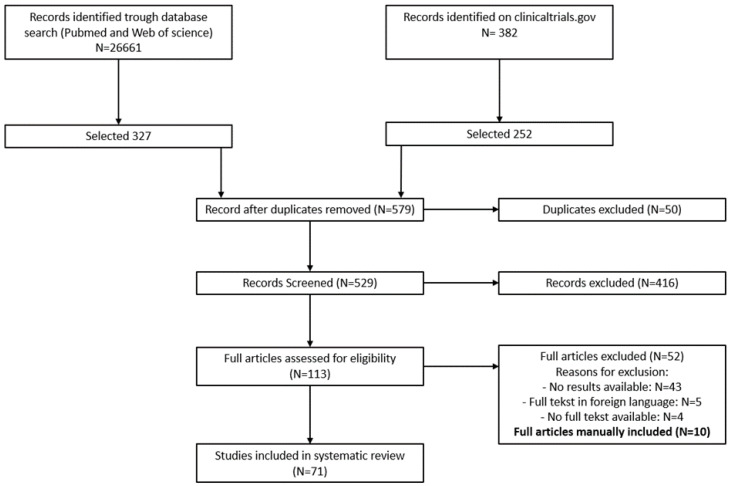
PRISMA flow diagram.

**Figure 2 biomolecules-12-00340-f002:**
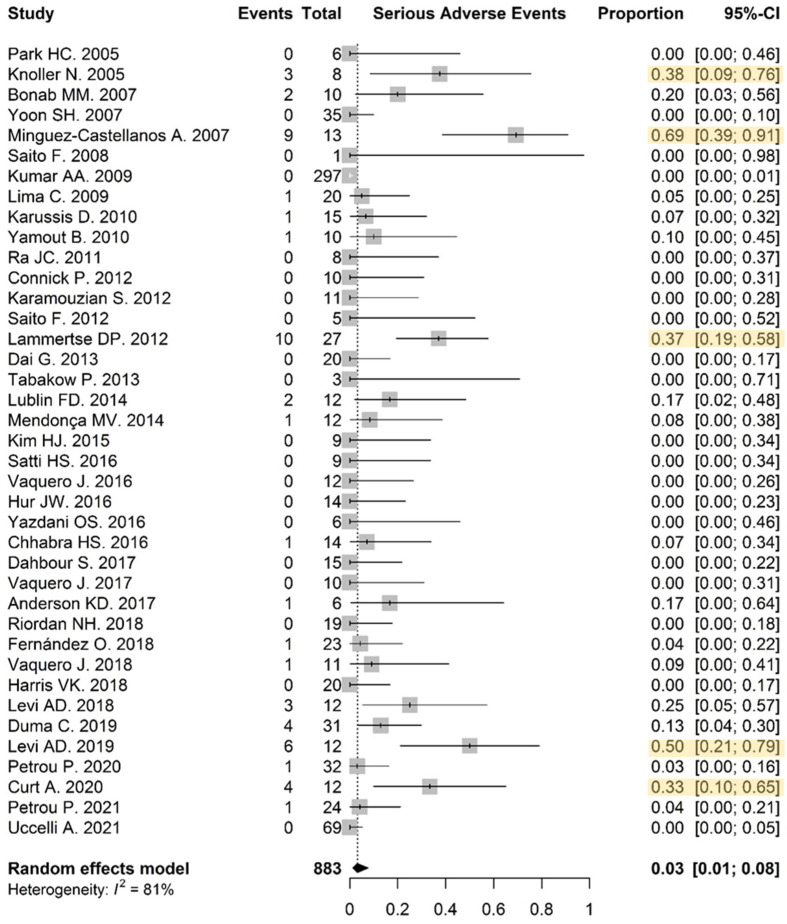
Safety—patients experiencing SAEs in the intervention group: the proportion was visualized for each study by the middle of the grey boxes including the 95% confidence interval indicated by the horizontal lines. The overall frequency of SAE was low: 0.03 (95% CI: 0.01–0.08). Five studies, marked in yellow, showed a significantly higher incidence of SAE.

**Figure 3 biomolecules-12-00340-f003:**
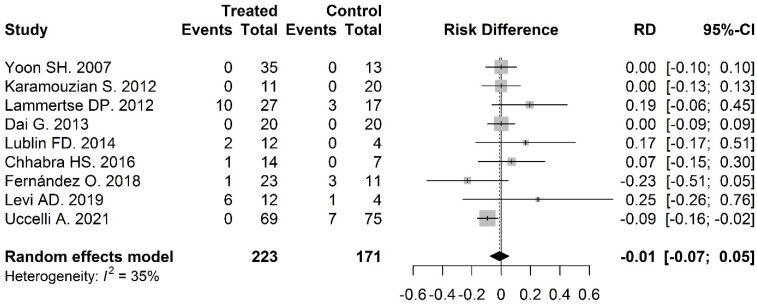
Safety—risk difference in SAE incidence between intervention and control arm: the risk difference for SAE in intervention groups versus control groups was −0.01 (95% CI: −0.07–0.05, *p* = 0.73).

**Figure 4 biomolecules-12-00340-f004:**
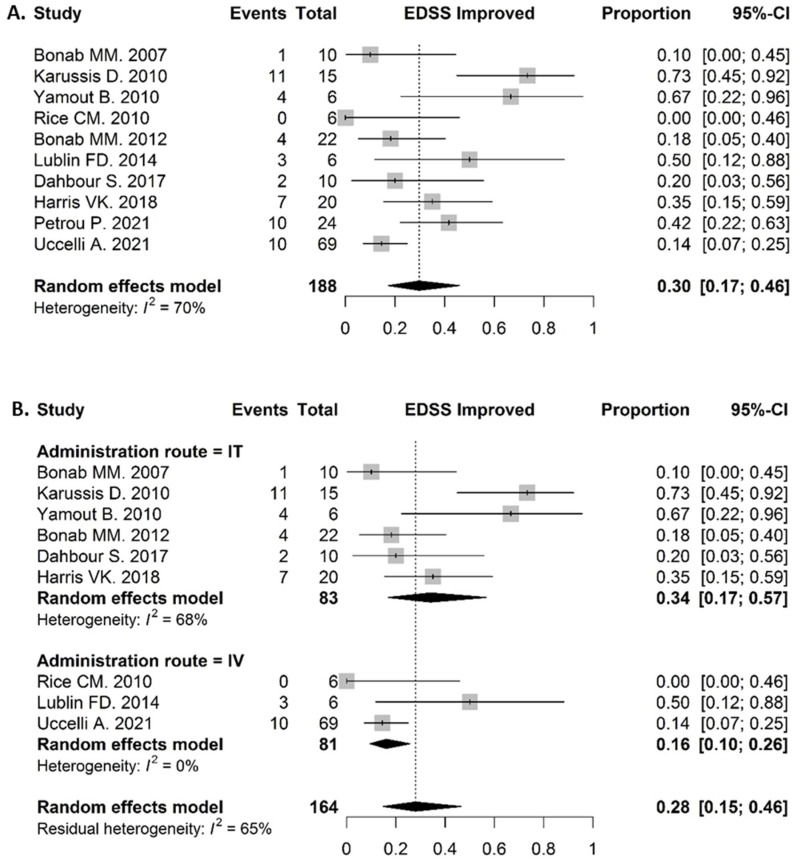
Efficacy. (**A**) Clinical response in MS patients: Proportion of patients with improved EDSS status: 0.30 (95% CI: 0.17–0.46). (**B**) Comparison in clinical response by administration route in MS patients: the proportion of patients who improved their EDSS status in the IT group was 0.34 (95% CI: 0.17–0.57), in the IV group 0.16 (95% CI: 0.10–0.26). The subgroup differences were not significant, *χ*^2^ = 3.11 and *p* = 0.078.

**Figure 5 biomolecules-12-00340-f005:**
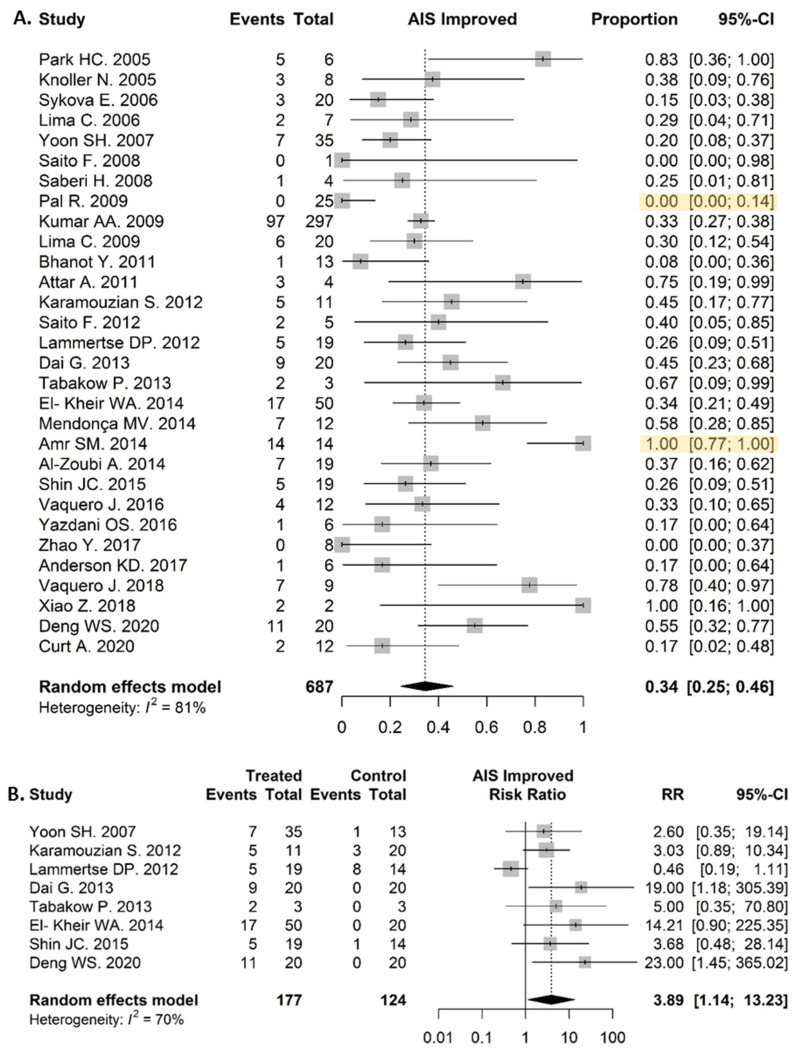
Efficacy. (**A**) Clinical response in SCI patients: 229/687 patients improved their SCI grade, resulting in a proportion of 0.35 (95% CI: 0.25–0.46). (**B**) Comparison of clinical improvement in SCI patients between treatment and control group: the proportion in the experimental group was remarkably higher (0.35 (95% CI: 0.27–0.44)) than in the control group (0.04 (95% CI: 0.01–0.22)). The calculated risk ratio was 3.89 (95% CI: 1.14–13.23) and *p* = 0.030.

**Figure 6 biomolecules-12-00340-f006:**
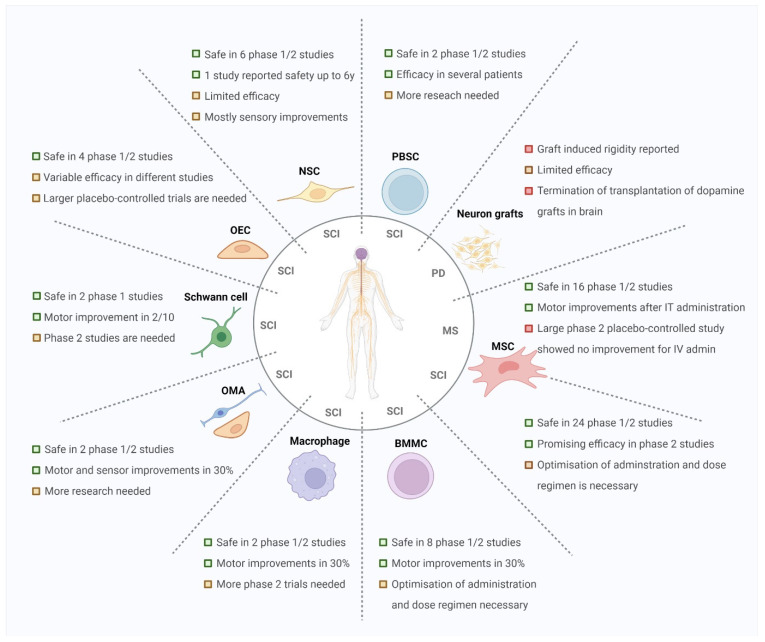
Overview of the current status of the different types of cell-based therapies in the clinic and their most common use in the different neurodegenerative diseases. Abbreviations: spinal cord injury (SCI), multiple sclerosis (MS), Parkinson’s disease (PD), intrathecal (IT), intravenous (IV), mesenchymal stromal cell (MSC), bone-marrow mononuclear cells (BMMC), olfactory mucosal autograft (OMA), olfactory ensheathing cell (OEC), neuronal stem cell (NSC), peripheral blood stem cell (PBSC).

**Table 1 biomolecules-12-00340-t001:** Overview of the 71 included studies. Abbreviations: mesenchymal stem cell (MSC), bone-marrow mononuclear cell (BMMC), olfactory mucosal autograft (OMA), neuronal stem cell (NSC), olfactory ensheathing cell (OEC), peripheral blood stem cell (PBSC), intravenous (IV), intrathecal (IT), intra-arterial (IA)relapse-remitting (RR), secondary-progressive (SP), primary-progressive (PP), relapsing-progressive (RP), acute (A), sub-acute (SA) chronic (C), varying subtypes (V), treatment (T), control (Co).

Author + Year	Cell Type	Administration Route	Disease	#Patients	%Improved
Riordan NH. 2018	MSC (UC)	IV	MS (RR,PP,SP)	T: 17	NA
Fernández O. 2018	MSC (Ad)	IV	MS (SP)	T: 19 Co: 10	T: NA Co: NA
Karussis D. 2010	MSC (BM)	IT	MS (V)	T: 15	73
Harris VK. 2018	MSC (BM)	IT	MS (PP,SP)	T: 20	35
Xiao Z. 2018	MSC (UC)	Injury Site	SCI (A)	T: 2	100
Ra JC. 2011	MSC (Ad)	IV	SCI (C)	T: 8	13
El-Kheir WA. 2014	MSC (BM)	IT	SCI (C)	T: 50 Co: 20	T: 34 Co: 0
Bonab MM. 2012	MSC (BM)	IT	MS (SP,PR)	T: 22	18
Dahbour S. 2017	MSC (Ad)	IT	MS (SP,RR)	T: 10	20
Lublin FD. 2014	MSC (PD)	IV	MS (RR,SP)	T: 12 Co: 4	T: 42 Co: 25
Llufriu S. 2014	MSC (BM)	IV	MS (RR)	T: 8	NA
Mendonça MV. 2014	MSC (BM)	Injury Site	SCI (C)	T: 12	58
Connick P. 2012	MSC (BM)	IV	MS (SP)	T: 10	NA
Cheng H. 2014	MSC (UC)	IT	SCI (C)	T: 10	NA
Li JF. 2014	MSC (UC)	IV	MS (RR,SP)	T: 13 Co: 10	T: NA Co: NA
Satti HS. 2016	MSC (BM)	IT	SCI (C)	T: 9	NA
Bonab MM. 2007	MSC (BM)	IT	MS (SP,PP)	T: 10	10
Dai G. 2013	MSC (BM)	Injury Site	SCI (C)	T: 20 Co: 20	45 0
Karamouzian S. 2012	MSC (BM)	IT	SCI (A,SA)	T: 11 Co: 20	T: 45 Co: 15
Vaquero J. 2016	MSC (BM)	IT + Injury site	SCI (C)	T: 12	33
Vaquero J. 2017	MSC (BM)	IT	SCI (C)	T: 10	NA
Vaquero J. 2018	MSC (BM)	IT	SCI (V)	T: 9	79
Oh SK. 2015	MSC (BM)	Injury site	SCI (C)	T: 16	NA
Hur JW. 2016	MSC (Ad)	IT	SCI (C)	T: 14	NA
Yamout B. 2010	MSC (BM)	IT	MS (SP,RR)	T: 7	57
Saito F. 2012	MSC (BM)	IT	SCI (A)	T: 5	40
Venktataramana NK. 2010	MSC (BM)	Surgery	PD	T: 7	43
Pal R. 2009	MSC (BM)	IT	SCI (C)	T: 25	0
Saito F. 2008	MSC (BM)	IT	SCI (A)	T: 1	0
Duma C. 2019	MSC (Ad)	CSF	SCI,MS,PD,AD	T: 31	NA
Kim HJ. 2015	MSC (UC)	Surgery	AD	T: 9	0
Petrou P. 2020	MSC (BM)	IT or IV	MS (SP,PP)	T: 32 Co: 16	T: NA Co: NA
Petrou P. 2021	MSC (BM)	IT and/or IV	MS (SP,PP)	T: 24	42
Uccelli A. 2021	MSC (BM)	IV	MS (RR,SP,PP)	T: 69 Co: 75	T: 15 Co: 14
Bhanot Y. 2011	MSC (BM)	IT + Injury site	SCI (C)	T: 13	8
Yazdani OS. 2016	MSC (BM) + SC	IT	SCI (C)	T: 6	17
Zhao Y. 2017	MSC (UC)	Injury site	SCI (C)	T: 8	0
Deng WS. 2020	MSC (UC)	Injury site	SCI (A)	T: 20 Co: 20	T: 55 Co: 0
Amr SM. 2014	MSC (BM)	Injury site	SCI (C)	T: 14	100
Rong L. 2020	MSC (UC)	IT	SCI (C)	T: 24	NA
Moviglia GA. 2006	MSC (BM)	Injury site	SCI (C)	T: 2	NA
Carstens M. 2020	MSC (Ad)	Facial	PD	T: 2	NA
Rice CM. 2010	BMMC	IV	MS (RP)	T: 6	0
Chhabra HS. 2016	BMMC	IT + Injury site	SCI (A)	T: 14 Co: 7	NA NA
Kumar AA. 2009	BMMC	IT	SCI (C)	T: 297	33
Yoon SH. 2007	BMMC	Injury site	SCI (V)	T: 35 Co: 13	T: 20 Co: 8
Chernykh ER. 2007	BMMC	IV + Injury site	SCI (C)	T: 18 Co: 18	NA NA
Syková E. 2006	BMMC	IV + IA	SCI (V)	T: 20	15
Park HC. 2005	BMMC	Injury site	SCI (A)	T: 6	83
Attar A. 2011	BMMC	Injury site	SCI (A)	T: 4	75
Lima C. 2006	OMA	Injury site	SCI (C)	T: 7	29
Lima C. 2009	OMA	Injury site	SCI (C)	T: 20	30
Curtis E. 2018	NSC	Injury site	SCI (C)	T: 4	NA
Levi AD. 2018	NSC	Injury site	SCI (C)	T: 39	NA
Levi AD. 2019	NSC	Injury site	SCI (C)	T: 12 Co: 4	NA NA
Shin JC. 2015	NSC	Injury site	SCI (V)	T: 19 Co: 15	T: 26 Co: 7
Curt A. 2020	NSC	Injury site	SCI (C)	T: 14	14
Mackay-Sim A. 2008	OEC	Injury site	SCI (C)	T: 3 Co: 3	T: 0 Co: 0
Féron F. 2005	OEC	Injury site	SCI (C)	T: 3 Co: 3	NA NA
Tabakow P. 2013	OEC	Injury site	SCI (C)	T: 3 Co: 3	T: 67 Co: 0
Wu J. 2012	OEC	Injury site	SCI (C)	T: 11	NA
Saberi H. 2008	Schwann	Injury site	SCI (C)	T: 4	25
Anderson KD. 2017	Schwann	Injury Site	SCI (C)	T: 6	17
Freed CR. 2001	Neuron	Surgery	PD	T: 20 Co: 20	T: NA Co: NA
Minguez-Castellanos 2007	Carotic Body	Surgery	PD	T: 12	83
Brundin P. 2000	Mesencephalic	Surgery	PD	T: 5	80
Al-Zoubi A. 2014	PBSC	IT + Injury site	SCI (C)	T: 19	37
Christante AF. 2009	PBSC	IA	SCI (C)	T: 39	NA
Lammertse DP. 2012	Macrophage	Injury site	SCI (A)	T: 19 Co: 14	T: 26 Co: 57
Knoller N. 2005	Macrophage	Injury site	SCI (A)	T: 8	38
Bakay RAE. 2004	Spheramine ©	Surgery	PD	T: 6	100

## Data Availability

Not applicable.

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
