# Peer review of "Are Cell-Based Therapies Safe and Effective in the Treatment of Neurodegenerative Diseases? A Systematic Review with Meta-Analysis"

_biomolecules, 2022, doi:10.3390/biom12020340_

Round 1
Reviewer 1 Report
Manuscript ID: biomolecules-1571650
Title: Are cell-based therapies the answer to neurodegenerative diseases? A systematic review with meta-analysis
Authors: Authors: Jasper Van den Bos, Yousra El Ouaamari, Kristien Wouters, Nathalie
Cools, Inez Wens
This review paper summarizes the results of long-term attempts to develop and use cellular technologies for the treatment of neurological diseases such as: multiple sclerosis, spinal cord injury, Parkinson's disease, and Alzheimer's disease. To do this, the authors analyzed dozens of studies in which different cells - mesenchymal, neuronal, stem, etc., were implanted in various parts of the animal nervous system when modeling the diseases. In addition, the authors summarize the results of clinical trials of the therapeutic efficacy of cellular technologies that have previously been tested in animal models of diseases. Analyzing a large amount of data, the authors conclude that clinical trials prove the safety and nonspecific neuroprotective effects of implanted cells. However, the authors emphasize that numerous clinical studies aimed at proving that implanted cells have a specific therapeutic effect, for example, compensating for a neurotransmitter deficiency in neurodegenerative diseases, have not yet been successful. Nevertheless the authors do not lose optimism, believing that the problem of developing cellular technologies for specific therapy of neurological diseases can be solved by searching for the optimal regime of their application - "dose", method of implantation, etc.
Thus, the paper under review makes it possible to understand the current state of development of cell technologies and the prospects for their use for the treatment of neurological diseases. The authors of the review do not doubt the success in solving this problem, although this will take some time. On the contrary, according to the reviewer opinion, the prospect of successful development and use of cellular technologies for the treatment of neurological patients in not too distant future is not very realistic (for more details, see below).
Major points
A serious drawback of the paper under review is the lack of analysis of the methodology and experience of the first clinical trials of the use of cell therapy for the treatment of Parkinson's disease in the 90s, coordinated by A. Björklund. These clinical trials were carried out as a multicenter project ("Network for European CNS transplantation and restoration - NECTAR") with the participation of the EU countries and the Russian Federation. The basis for starting clinical studies of transplantation of human embryonic dopaminergic neurons into the striatum of patients with Parkinson's disease (PD) was the evidence that this approach leads to a complete restoration of motor behavior in rodents when modeling this disease. Further clinical studies in various countries participating in the project were carried out according to the same protocol, including (i) a comprehensive examination of patients before surgery, (ii) preparation of cellular material and stereotactic injection into the patient's brain, (iii) regular long-term comprehensive examination of patients for a long time after surgery, (iv) analysis of pathological material obtained after death of some patients due to a cause independent of PD and surgery. As a result of clinical studies, the following conclusions were made: (i) transplanted neurons survive in the brain of the patient and innervate his striatum; (ii) there is an improvement in motor activity, but only in some operated patients and for a rather limited time. On this basis, clinical trials were stopped in the early 2000th.
As the authors of the reviewed paper point out, over the past two decades, a lot of experience has been accumulated in improving cellular technologies for the treatment of neurological diseases and evaluating their therapeutic efficacy in animal models and patients. Several types of cells - embryonic neurons, genetically engineered cells of neuronal and non-neuronal origin, stem cells and their derivatives, were used in these studies. However, the principal results of these studies are similar to the results obtained in the frame of the NECTAR project. Indeed, using cellular technologies, researchers usually obtain perfect specific therapeutic effects in animal models but not in treated patients. Based on the above, the question arises whether the negative experience of using cellular technologies for the treatment of patients with neurological diseases is a result of the insufficient development of the technique for preparing cells and their implantation, as the authors of the reviewed paper state, or a consequence of a biological taboo on the functional integration of implanted cells into the human brain. In the opinion of the reviewer, a special discussion of this issue will draw the attention of researchers to the study of the mechanisms of the functional incompatibility of the implanted cells with the human brain supposed by the reviewer.
Thus, the article can be reviewed again after a major revision.
Reviewer 2 Report
In this systematic review the authors discuss the potential effectiveness of cell-based therapies to treat neurodegenerative diseases. The manuscript is focused in multiple sclerosis, spinal cord injury, Parkinson's disease and Alzheimer's disease and the effects of mainly MSCs infusions to treat these diseases. Overall, the manuscript is interesting, well organized and written. However, I have some comments that should be addressed in order to improve the quality of the work.
1- I think that after reading the paper, the question raised in the title still remains unanswered. It is clear that the heterogeneity of the performed clinical trials, in terms of number and sources of cells, autologous versus allogeneic, route, etc is a key factor determining the variability of the obtained outcomes but in spite of this, there are two observations that are pretty clear: MSCs administration is safe, and, yes, at least in part, it seems to be effective. Many questions remain to be elucidated such as the convenience of using autologous (coming from adult patients with a declined functionality of their MSCs) or allogeneic MSCs from younger donors and with increased functional capabilities, among them their paracrine properties. Moreover, these paracrine properties are thought to be the main mechanism driving the therapeutic potential of infused MSCs. Unfortunately, these effects are transitory and therefore, the strategy of administering reiterative infusions is being considered to maintain the benefitial effects of MSCs. In fact, although far from neurodegenerative diseases, this has been already tested in pediatric patients with Osteogenesis Imperfecta, with encouraging results.
2. Considering the comment 1, I suggest to change the title for this one (or a similar one): Effectiveness of cell-based therapies for neurodegenerative diseases: A systematic review and meta-analysis.
3. Considering the comment 1, I think that the mentioned observations should be more clearly exposed in the discussion. An illustration should be interesting and would ease to assimilate the take home message of the paper.
4. The Conclusions are literally a copy of the Future perspectives. In their present form they do not conclude the effectiveness of cell therapies to treat neurodegenerative diseases. On the contrary, they mention new strategies and approaches that are not within the scope of this review, and which have been barely introduced (scaffolds, paracrine properties). If the authors want to mention these approaches, which on the other hand are totally relevant and would enrich the paper, they should write another section introducing, mentioning and discussing them. In fact, the end of the manuscript should be more concise taking into account the discussed reviewd outcomes of cell therapies for neurodegenerative diseases along the paper, and not an open question about neurotrophins.
Round 2
Reviewer 1 Report
The review paper has been substantially revised at the request of the reviewer. However, manifestations of inadequate optimism of the authors regarding the use of cellular technologies for the treatment of patients with neurodegenerative diseases (NDD) still dominate. Already from the title of the paper follows the authors' conviction in the safety and effectiveness of this approach to the treatment of NDD. The title should preferably be presented as a polemical question like “Are cellular technologies safe and effective in the treatment of NDD?”
At the very beginning of the paper (abstract), the authors emphasize that the analysis is limited to the literature of the last 20 years, although the first fundamental clinical trials of neurotransplantation in Parkinson's disease were conducted in the 90s and early 2000s. Moreover, the results of preclinical and clinical studies of recent years do not fundamentally differ from those obtained in the NECTAR program, which showed an excellent therapeutic effect in animal models and the absence of a stable specific therapeutic effect in most patients. The authors believe that to achieve success, it is sufficient to optimize the conditions for implantation: "optimization is still needed regarding dose, frequency and administration route". However, it is not discussed at all that the failure of clinical trials, especially in the case of such NDD as PD, may be due to biological restrictions (tabu), such as the inability to reproduce specific synaptic connections. In the history of regenerative medicine, an example of biological restriction was represented by immunorejection. Only after discovery of immunorejection and the production of immunosuppressants the transplantation of the internal organs became feasible. Given from the above, it is desirable to soften throughout the paper such categorical statements of the authors as: “Over the past two decades, significant advances have been made in the field of regenerative medicine…”.
Reviewer 2 Report
The authors have satisfactorily addressed the previous suggestions. Congratulations for the new added figure.
Author Response
No further comments were needed.
Round 3
Reviewer 1 Report
The authors answered all the questions and comments of the reviewer. Article is recommended for publication